# Visible-light promoted late-stage chlorination and bromination of quinones and (hetero)arenes utilizing aqueous HCl or HBr as halogen donors

Yangyang Zhang, Jinglian Nong & Yaxin Wang ✉

Late-stage C–H chlorination and bromination of bioactive scaffolds is significant for drug discovery, as carbon–halogen bonds can effectively modulate biological activity, metabolic stability, and physicochemical profiles. Here, we develop an atom-economical, radical-mediated protocol that synthesizes high-value chlorinated or brominated quinones and (hetero)arenes. The system employs inexpensive, low-molecular-weight HCl or HBr as the halogen source, commercially available $NaNO_2$, and blue light irradiation, without any photocatalyst or metal catalyst. The mild reaction conditions, ready availability of reagents, excellent functional-group tolerance, high regioselectivity, and facile scalability under continuous-flow operation collectively render this approach a practical and efficient protocol for the late-stage aromatic $C(sp^2)$–H chlorination and bromination of complex drugs and natural products. Mechanistic investigations reveal that nitrosyl halides, generated in situ by reaction of $NaNO_2$ and aqueous HCl or HBr, undergo photo-promoted homolysis to produce the corresponding halogen radicals that selectively initiate the radical halogenation of quinones and (hetero)arenes.

Late-stage C–H functionalization of bioactive molecules is a powerful strategy for creating a diverse molecule library to explore structure-activity relationships and optimize druggability[1–6]. Among the various reported late-stage functionalization reactions, C–H chlorination and bromination have proven to be particularly useful, due to the unique properties of carbon-halogen bond (C–Cl and C–Br) on modulating biological activity, metabolic stability, and physicochemical properties of drugs and natural products[7–14]. Additionally, aromatic $C(sp^2)$–Cl and $C(sp^2)$–Br bonds are prevalent in a wide range of high-value compounds, including fine chemicals[15–17], approved pharmaceuticals[18–20], natural products[21], agrochemicals[22,23], and functional materials (Fig. 1A). These bonds are also extensively utilized in coupling reactions, highlighting their significance in synthetic chemistry[24–26]. Therefore, developing aromatic $C(sp^2)$–H halogenation methods with better reactivity, selectivity and biocompatibility is necessary.

Traditional electrophilic aromatic substitution ($S_EAr$) methods for aromatic $C(sp^2)$–H chlorination or bromination typically employ highly reactive halogenating reagents (such as $Cl_2$, $Br_2$, or $SOCl_2$), Lewis acids (such as $FeCl_3$, $FeBr_3$, or $AlCl_3$), and harsh reaction conditions. These approaches often suffer from low chemoselectivity and regioselectivity, as well as poor compatibility with functional groups (Fig. 1B)[27,28]. To address these

limitations, various alternative halogenating reagents have been developed in recent years[29–38]. Although these new reagents have significantly improved the efficiency and regioselectivity of aromatic halogenation reactions, some of them exhibit lower atom economy, are prepared through complex synthetic processes, or suffer from instability issues (Fig. 1B)[29–38]. Given these challenges, the development of selective aromatic $C(sp^2)$–H halogenation methods that employ practical, commercially available, and inexpensive halogenating reagents—such as aqueous HX (X = Cl, Br), halogen-containing inorganic salts, or halogenated solvents—is urgently needed, as this would improve atom economy and reduce reaction costs.

With the recent advances in photochemistry[39–49], photo-promoted aromatic $C(sp^2)$–H halogenation reactions have developed rapidly (Fig. 1C)[7–14,50–63]. To date, visible-light-promoted $C(sp^2)$–H halogenation of (hetero)arenes has been dominated by two paradigms. The first relies on N–Cl or N–Br reagents (such as NCS, NBS, PhthN–Cl, PhthN–Br, NCSacc, TCCA, DCDMH and Palau'chlor reagent) in combination with a photocatalyst to deliver selective aromatic $C(sp^2)$–H chlorinated or brominated products. This manifold requires a photocatalyst and inevitably generates stoichiometric by-products, thereby compromising the atom economy. The second manifold employs aqueous HCl or HBr, or inorganic halide salts as

College of Pharmacy, Nanjing University of Chinese Medicine, Nanjing, China. ✉e-mail: 300500@njucm.edu.cn

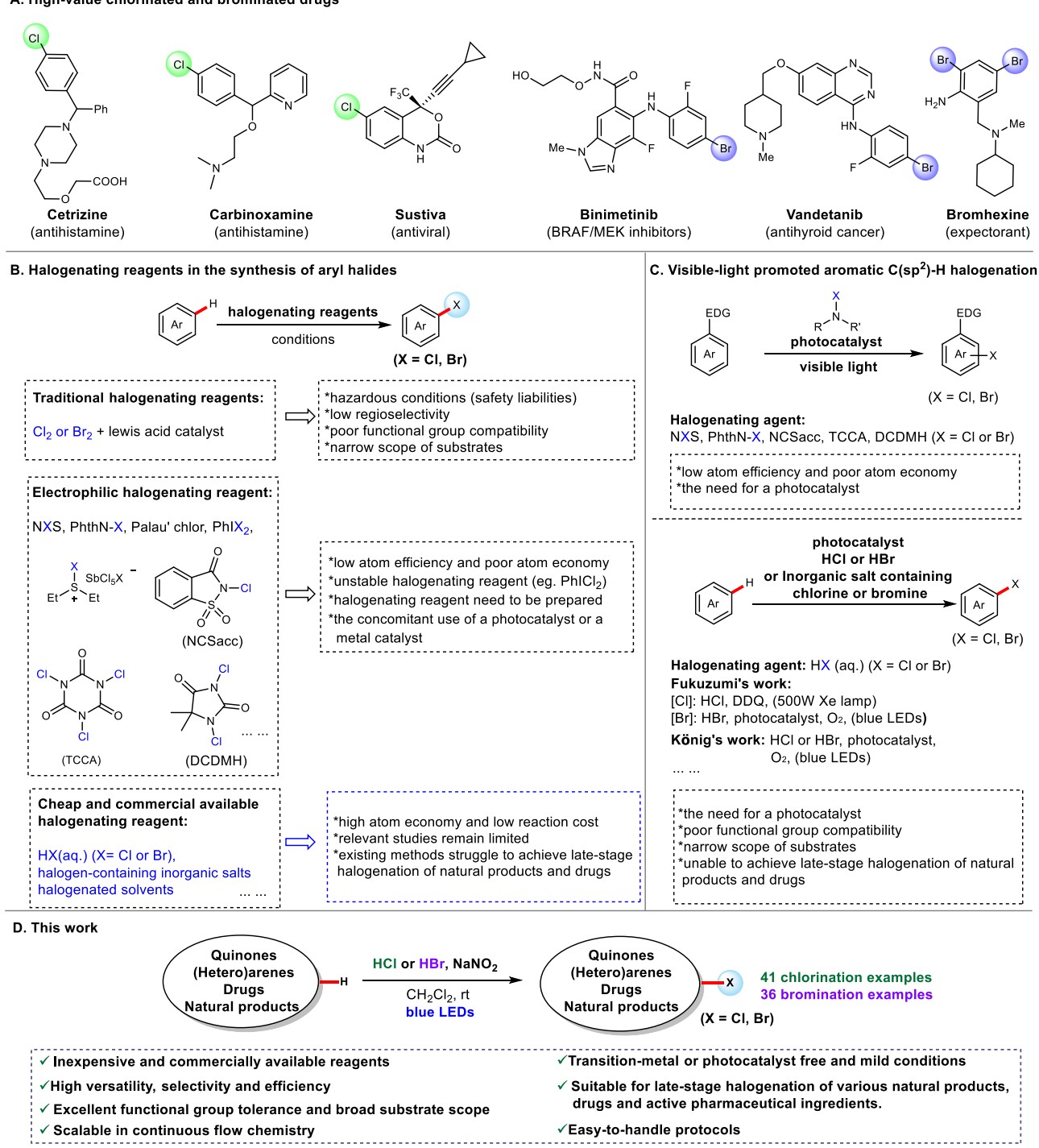

**Fig. 1 | Selective examples of halogenated drugs and halogenation reactions.**
**A** Examples of chlorinated and brominated drugs. **B** Halogenating reagents in the synthesis of aryl halides. **C** Visible-light promoted aromatic C(sp²)–H chlorination and bromination. **D** Versatile, tunable method for NaNO₂/HX–mediated radical chlorination or bromination of quinones, (hetero)arenes, drugs and natural products.

the halogen source; a strong chemical oxidant or strongly oxidizing photocatalyst is then required to generate the corresponding halogen radical that adds to the (hetero)arene. These strongly oxidative conditions erode functional-group tolerance and restrict substrate scope, rendering late-stage C(sp²)–H halogenation of complex natural products or drug scaffolds difficult. For example, The Fukuzumi group reported the first visible-light-induced bromination of MeO-substituted aromatic hydrocarbons and thiophenes using 9-mesityl-10-methylacridinium perchlorate as a photocatalyst and aqueous HBr as the brominating reagent under an oxygen atmosphere[54]. Subsequently, they also developed a visible-light-promoted

chlorination protocol for MeO-substituted arenes and phenols, utilizing HCl as the chlorine source[55]. Similarly, the König group introduced a method that integrates aqueous HCl into photocatalysis to achieve selective chlorination of simple electron-rich arenes under visible light irradiation[57,58]. Despite significant advances in visible-light-induced aromatic C–H halogenation using HX as the halogen source, most protocols still require harsh illumination conditions, exogenous photosensitizers and are largely limited to simple, low-molecular-weight substrates with narrow functional-group tolerance. To the best of our knowledge, there is still a lack of practical approaches for late-stage chlorination and bromination of natural products

**Table 1 | Optimization of conditions for chlorination and bromination of drug molecule naproxen[a]**

| Entry | Reagents (equiv), HCl (37% aq.) or HBr (40% aq.) (uL) | 1% or 2% | 1' % |
|---|---|---|---|
| 1 | NaNO$_2$ (0.2), HCl (12 μL, 1.4) | 26 (**1**) | <10 |
| 2 | NaNO$_2$ (0.5), HCl (30 μL, 3.6) | 60 (**1**) | <10 |
| 3 | NaNO$_2$ (1.0), HCl (60 μL, 7.2) | 86 (75[b]) (**1**) | <10 |
| 4 | NaNO$_2$ (1.0), HCl (50 μL, 6.0) | 85 (75[b]) (**1**) | <10 |
| 5 | NaNO$_2$ (1.0), HCl (30 μL, 3.6) | 29 (**1**) | <10 |
| 6 | NaNO$_2$ (1.0), HCl (10 μL, 1.2) | 13 (**1**) | <10 |
| 7 | NaNO$_2$ (2.0), HCl (50 μL, 6.0) | 51 (**1**) | 24 |
| 8[c] | NaNO$_2$ (1.0), HCl (50 μL), in darkness | <10 (**1**) | 11 |
| 9 | HCl (50), 12 h | 0 (**1**) | 0 |
| 10 | NaNO$_2$ (1.0), HCl (50 μL), N$_2$ | 17 (**1**) | <10 |
| 11 | NaNO$_2$ (1.0), HCl (50 μL), O$_2$ | 67 (**1**) | 18 |
| 12 | NCS (5.0) | 10% (**1**) | 0 |
| 13 | DCDMH (5.0) | 60% (**1**) | 0 |
| 14 | PhICl$_2$ (5.0) | 68% (**1**) | 0 |
| 15 | Palau'chlor (5.0) | 56% (**1**) | 0 |
| 16 | NaNO$_2$ (1.0), HBr (50 μL, 3.5) | 90 (80[b]) (**2**) | <10 |

[a]All reactions were carried out on a 0.1 mmol scale in solvent dichloromethane (DCM) (1 mL) at room temperature (24 °C) under blue LEDs (460–470 nm) (24 W) irradiation for 10 h, and yields are based on $^1$H NMR analysis on a 0.1 mmol scale reaction mixture. HCl (37% aqueous) and HBr (40% aqueous).
[b]Isolated yield.
[c]In darkness.

and drugs using a commercially accessible aqueous HCl or HBr as halogen source.

Based on these prior studies and our own work on visible-light-promoted reactions[64–70], we herein report a NaNO$_2$/HCl-mediated radical strategy for the selective C(sp$^2$)–H chlorination and bromination of quinones, (hetero)arenes, natural products, drugs, and active pharmaceutical ingredients. This strategy operates under visible-light irradiation at room temperature without the need for a photo- or metal-catalyst. Furthermore, by simply replacing aqueous HCl with aqueous HBr, this system can be conveniently modulated to achieve late-stage C(sp$^2$)–H bromination of complex substrates. Moreover, these halogenated reactions can be readily scaled up via continuous-flow technology (Fig. 1D).

## Results and discussion
### Screening conditions
Our research in radical C–H late-stage halogenation chemistry is motivated by our interest in the synthesis and biological activity studies of drugs. To

this end, we initiated our investigation of selective C(sp$^2$)–H chlorination using the challenging drug molecule naproxen (**1s**) as the substrate and inexpensive, readily available aqueous HCl as the chlorine donor under blue LEDs irradiation (Table 1 and Supplementary Data 2).

We were pleased to find that the reaction of naproxen (**1s**) (0.1 mmol) with 37% aqueous HCl (12 μL) and 0.2 equivalents of NaNO$_2$ in dichloromethane (DCM) under blue light irradiation at room temperature (24 °C) afforded a 26% yield of the selective C(sp$^2$)–H chlorination product (**1**) (Table 1, entry 1). Moreover, while keeping the equivalent ratio of aqueous HCl and NaNO$_2$ constant, increasing the equivalents of NaNO$_2$ and HCl led to a further increase in the yield of chlorinated naproxen (compare entries 1, 2, and 3). Specifically, using 1.0 equivalent of NaNO$_2$ and aqueous HCl (60 μL) resulted in a 75% isolated yield of product **1** (86% yield based on crude $^1$H-NMR spectroscopy) (entry 3). Furthermore, reducing the amount of aqueous HCl to 50 μL maintained the yield at a similar level (entry 4). However, further reduction in the amount of aqueous HCl led to a decrease in the

chlorination yield (entries 5 and 6). When we used 2.0 equivalents of NaNO$_2$ and 50 μL of aqueous HCl, both chlorinated product **1** (51% yield) and nitrated product **1′** (24% yield) were obtained (entry 7). Choice of light source clearly has an impact on the yield of reaction of **1** (see the Supporting Information for details, Supplementary Table S1, entry 8–11). Reaction under red LEDs (620–630 nm) or a UV lamp (365 nm) irradiation gave only a small amount of chlorinated naproxen (**1**), and irradiation of the reaction mixture with a white compact fluorescent lamp or a green LEDs (510–520 nm) of the same power gave moderate yields of chlorinated product (**1**). The best result was obtained with blue LEDs (460–470 nm) (entry 4). Control experiments were carried out to prove that both NaNO$_2$ and visible light were necessary for the formation of chloride (**1**) (entries 8 and 9). To probe the possible role of atmospheric oxygen, we conducted the C–H chlorination of Naproxen (**1s**) under rigorously controlled atmospheres (air, O$_2$, and N$_2$) (entries 4, 10 and 11). The results show that the chlorination reaction under the air gave the best yield of dichlorination product **1**, and the reaction under the O$_2$ atmosphere afforded the chlorination product **1** in 67% yield and the nitration product **1′** in 18% yield. However, under an N$_2$ atmosphere, the reaction afforded the C(sp$^2$)–H chlorinated product **1** in 17% yield, with trace nitration product **1'** detected, and 70% of the starting material left unreacted. These results showed that ambient air exerts a beneficial effect on the reaction, whereas an O$_2$-rich atmosphere readily oxidizes NO to NO$_2$, thereby diverting the pathway toward nitration and increasing the yield of the nitro product **1'**. Moreover, under blue-light irradiation, substituting aqueous HCl and sodium nitrite with alternative chlorinating reagents—such as NCS, DSDMH, PhICl$_2$ or Palau'chlor—afforded the target chlorinated products only in moderate to low yields (entries 12–15). As anticipated, this system can be further modulated to form selective C(sp$^2$)–H brominated product **2** in a nearly quantitative yield by replacing aqueous HCl with aqueous HBr (entry 16).

With two optimized sets of reaction conditions in hand (Table 1, entries 4 and 16), we evaluated the substrate scope of the selective C(sp$^2$)–H chlorination and bromination reactions.

### Visible-light-promoted late-stage chlorination of quinones, (hetero)arenes, natural products and drugs

As shown in Scheme 1, a diverse array of quinolones, arenes, and heteroarenes underwent NaNO$_2$/HCl-mediated chlorination under blue light irradiation at room temperature, affording the corresponding C(sp$^2$)–H chlorination products (**2–33**) with excellent regioselectivity. A variety of functional groups, including halogen atoms (F, Cl, Br), ethers, esters, amides, tertiary amines, carboxylic acids, cyano groups, and sulfonyl groups, were well tolerated under these conditions. Specifically, 1,4-benzoquinone delivered 56% of mono-chlorinated and 32% of di-chlorinated products (**3 and 3'**) under the standard chlorination conditions. However, upon precisely reducing the aqueous HCl loading to 3.0 equivalents, mono-chlorinated benzoquinone (**3**) is obtained as the sole isolable product in 87% isolated yield. 1,4-Benzoquinones containing electron-withdrawing chloride substituents or electron-donating groups (Me- and MeO-) all gave corresponding mono-chlorinated products in high yields (**4–10**). Among the tested naphthoquinones, increased steric hindrance of the naphthoquinones has a negative effect on the reaction yield (**11 vs 12**). Moreover, the electronegativity of substituents on naphthoquinones also has a slight impact on the yield of the chlorination, as evidenced by the contrasting results between 2-Chloro-1,4-naphthoquinone (**12**) and Menadione (**13**). 1,4-Anthraquinone also worked under the standard chlorination conditions, but resulted in relatively low yield of the corresponding dichlorination product (**14**). Likewise, by adjusting the aqueous HCl to 3.0 equiv we sought to obtain the mono-chlorinated 1,4-anthraquinone; regrettably, 1,4-anthraquinone afforded only a trace amount of the desired mono-chloro product (~3%), with 90% of the starting material being recovered. Additionally, various substituted electron-rich arenes can all be smoothly chlorinated with moderate to good yields and excellent regioselectivities (**15–24**). Notably, C–H chlorination of ethoxybenzene

exclusively affords the *para*-substituted chlorination product (**16**), with no *ortho*-chlorinated product observed, as a consequence of steric hindrance. Moreover, the tertiary amine of 1-phenylpiperidine remains intact, despite its oxidant sensitivity, likely due to protection by protonation under the acidic condition (**25**). Furthermore, heteroarenes such as pyrazole, indole, imidazo[1,2-a]pyridine, imidazo[1,2-a]pyrimidine, and thiophene derivatives were all efficiently chlorinated with high regioselectivity (**26–32**). Significantly, the electron-deficient heteroarene 2,6-dimethoxypyridine also performed in the chlorination system, although with a lower isolated yield, e.g., **33**. Regrettably, other electron-deficient (hetero)aromatic compounds—including quinoline, benzoic acid, methyl benzoate, and nitrobenzene—proved entirely unreactive under the optimized conditions (see the Supporting Information for details, Scheme S7).

The mild reaction conditions and excellent functional group compatibility enable this selective chlorination protocol to be readily applied to the late-stage chlorination of pharmaceuticals and natural products (Scheme 1, compounds **1** and **34–42**). Remarkably, a diverse range of drugs and drug fragments, including naproxen (**1**), leflunomide (**34**), apremilast (**35**), alogliptin derivative (**36**), and acetaminophen derivative (**37**), successfully underwent the selective aromatic C(sp$^2$)–H chlorination reaction, affording the desired chlorinated products in moderate to good yields. Moreover, the chlorinated analogs of caffeine (**38**), xanthotoxin (**39**), coumarin (**40–41**), and uracil (**42**) were each efficiently accessed in a single step using this protocol.

### Visible-light-promoted late-stage bromination of quinolones, (hetero)arenes, natural products and drugs

As shown in Scheme 2, the visible-light-promoted C(sp$^2$)–H bromination reactions of quinones, arenes, and heteroarenes with NaNO$_2$ and aqueous HBr as the bromine donor also demonstrated excellent efficiency, site-selectivity and functional group tolerance (**43–67**). Notably, in the bromination of 1,4-dimethoxybenzene and 1,3-dimethoxybenzene, our standard conditions afforded the dibrominated products **51** and **52** in excellent yields. However, by precisely reducing the aqueous HBr loading to 1.7 equivalents, both substrates provided the monobrominated product in 90% yield. As expected, the NaNO$_2$/HBr mediated bromination system was successfully employed for the late-stage bromination of pharmaceuticals and natural products (Schemes 2, **2** and **68–79**). Particularly, the system exhibited excellent tolerance towards acid-sensitive groups (e.g., ether, alkene), base-sensitive groups (e.g., amide, ester, carboxylic acid), and oxidant-sensitive groups (e.g., tertiary amine, N-heterocycle, aldehyde). Specifically, a range of drugs and drug fragments, such as naproxen, leflunomide, apremilast, an alogliptin derivative, and an acetaminophen derivative, underwent bromination under our optimized conditions, affording the desired products **2** and **68–71** in 48–91% yield with high regioselectivity. Moreover, the bromination of several natural products, including xanthotoxin (**72**), coumarin (**73 and 74**), uracil (**76**), veratrole (**76**), and vanillin (**77**), afforded the corresponding brominated products in moderate to high yields (56–90%). Given the importance of tyrosine and tryptophan in proteins and peptide drugs such as alarelin and oxytocin, we explored the late-stage bromination of these amino acids. Encouragingly, the C(sp$^2$)–H bromination of N-benzoyl tyrosine methyl ester and N-phthaloyl tryptophan pentenyl ester under standard visible-light-promoted conditions afforded the corresponding brominated products with good yields and excellent regionselectivity (**78 and 79**).

The simplicity, generality, and low cost of this NaNO$_2$/HX system suggest its potential utility for the derivatization of specialty chemicals and the development of novel bioactive compounds.

To further demonstrate the synthetic potential of our protocol, we conducted gram-scale continuous-flow reactions to synthesize chlorinated naproxen (**1**) and brominated naproxen (**2**), respectively[71]. Utilizing a flow rate of 1 mL min$^{-1}$ and a residence time of 10 hours, we successfully scaled up the chlorination and bromination reactions by a factor of 100 (to 10 mmol) with negligible impact on the reaction yields (Fig. 2).

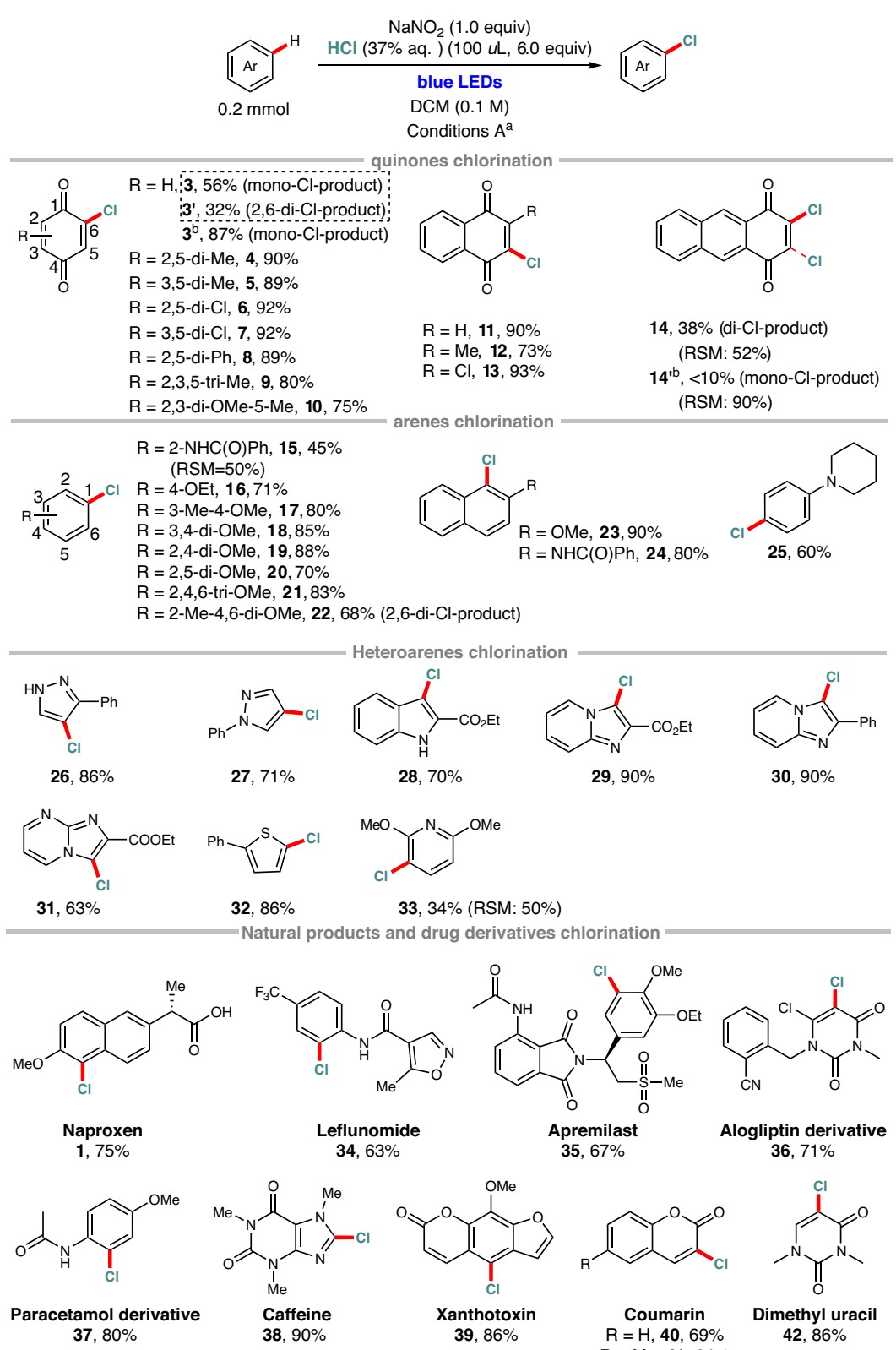

**Scheme 1** | Visible-light-promoted chlorination of quinones, (hetero)arenes, natural products and drugs. [a]Standard reaction conditions: substrate (0.2 mmol, 1.0 equiv), NaNO₂ (0.2 mmol, 1.0 equiv), and 37% aqueous HCl (100 μL, 6.0 equiv) in DCM (2 mL) were allowed to react at room temperature (24 °C) in air under irradiation with 24 W blue LEDs (460–470 nm). RSM, recovered starting material. [b]Reducing the aqueous HCl loading to 50 μL (3.0 equiv).

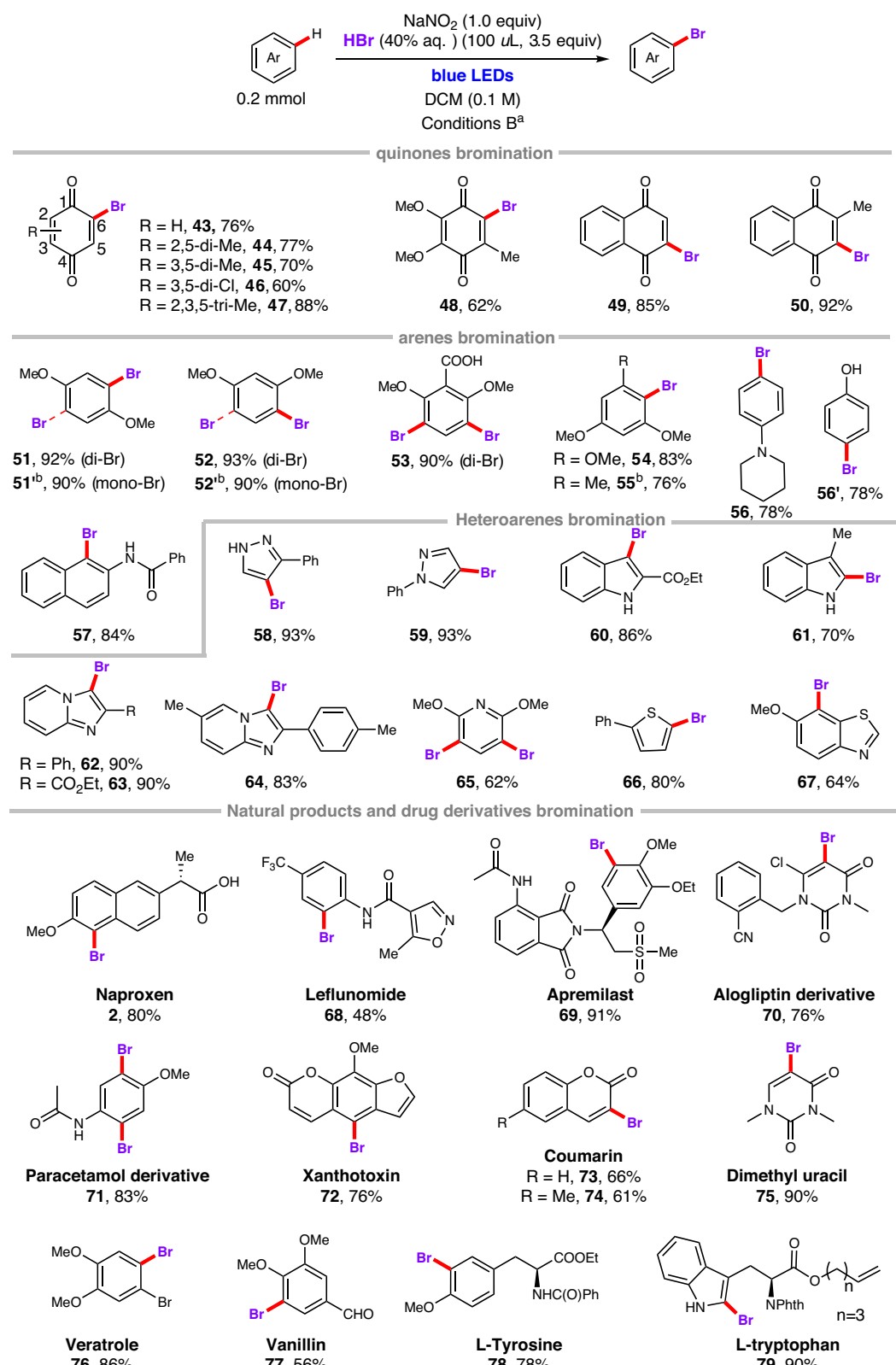

**Scheme 2 | Visible-light-promoted bromination of quinones, (hetero)arenes, natural products and drugs. **[a]**Standard reaction conditions: substrate (0.2 mmol, 1.0 equiv), NaNO$_2$ (0.2 mmol, 1.0 equiv), and 40% aqueous HBr (100 μL, 3.5 equiv) in DCM (2 mL) were allowed to react at room temperature (24 °C) in air under irradiation with 24 W blue LEDs (460–470 nm). RSM, recovered starting material. **[b]**Reducing the aqueous HBr loading to 50 μL (1.7 equiv).

**Fig. 2 | Synthetic applications. A** Gram-scale continuous-flow chlorination reaction of naproxen. **B** Gram-scale continuous-flow bromination reaction of naproxen.

**A) Gram-scale continuous flow aromatic C(sp²)-H chlorination of naproxen**

**B) Gram-scale continuous flow aromatic C(sp²)-H bromination of naproxen**

## Mechanistic studies

A series of experiments were conducted to elucidate the mechanism of the NaNO$_2$/HCl-mediated aromatic C(sp²)–H chlorination reaction (Fig. 3). The addition of the radical-trapping reagent diphenylethylene was found to inhibit the formation of chlorinated product (**1**) and to afford the chlorine radical-trapping product (**80**) in 38% yield (Fig. 3A). Moreover, the benzylic C(sp³)–H bond of 1,2-dimethoxy-4-methylbenzene can be activated by a chlorine radical through a hydrogen atom transfer (HAT) process, giving the corresponding chlorinated product (**81**) in 63% isolated yield (Fig. 3B). Collectively, the radical scavenger and the HAT experiment clearly demonstrate that chlorine radical is initially formed under the standard chlorination conditions. We next sought to confirm the radical pathway using two different radical clock substrates. Specifically, N-tosyl diallylamine (**82s**) furnished the 5-exo-trig cyclization product (**82**) in 67% yield, while cyclopropyl-substituted alkene (**83s**) underwent facile ring-opening upon initial chlorine radical addition, yielding the expected product (**83**) with 65% yield and excellent *E/Z* selectivity (Fig. 3C). Furthermore, the light-on/light-off experiment for the chlorination of dimethyluracil (**42**) demonstrated that although the reaction proceeds in the dark, the reaction rate is significantly diminished (Supplementary Data 3). This finding underscores the pronounced promoting effect of visible light on the reaction while simultaneously revealing that a radical-chain pathway is involved in the absence of illumination[72] (Fig. 3D) (see the Supporting Information for details, S39–42). Additionally, UV–vis experiments were carried out. A series of aliquots were removed from mixtures of NaNO$_2$ and HCl, transferred to a quartz tube, and analyzed directly. The UV–vis spectra of the mixtures showed an absorption band centered at 330 nm, 346 nm, 358 nm, 371 nm and 385 nm, which agrees well with photoinduced depletion products of ground-state nitrosyl chloride (ClNO)[73–75] (Fig. 3E).

Similar mechanistic experiments were conducted, which confirmed the formation of bromine radical under the standard bromination conditions and demonstrated the involvement of a radical mechanism in the bromination reactions (see the Supporting Information for details, S43–45).

Based on our mechanistic experiments and literature reports[76–78], we propose a plausible pathway for NaNO$_2$/HCl-mediated photochemical aromatic C(sp²)–H chlorination. The radical process starts with the formation of nitrous acid (HNO$_2$) by a reaction between NaNO$_2$ and HCl. Then, nucleophilic substitution reaction between **Int I** (protonated HNO$_2$) and aqueous HCl in situ generates chlorinating reagent nitrosyl chloride (Cl–N=O). Under light irradiation, nitrosyl chloride undergoes homolytic cleavage to generate chlorine radical and nitric oxide radical. Subsequently, the chlorine radical selectively adds to the aromatic π-bond of the substrate, forming a carbon-centered radical intermediate (**Int II**). **Int II** undergoes a single-electron transfer (SET) with nitrosyl chloride to generate carbocation intermediate (**Int III**). A trace amount of H$_2$O present in aqueous HCl then acts as a base to facilitate proton elimination of **Int III**, furnishing the aromatic C(sp²)–H chlorinated product; the concomitant restoration of aromaticity provides the principal thermodynamic driving force for this deprotonation event. Meanwhile, the one-electron-reduced nitrosyl chloride undergoes fragmentation to afford a chloride anion and nitric oxide. The latter is subsequently scavenged by air(O$_2$) and H$_2$O to regenerate nitrous acid, thereby re-entering the cycle. In the light-on/light-off experiment, the clear self-sustained component points toward a radical chain process being involved (see the Supporting Information for details, S39–43)[72].

## Conclusion

In summary, we have developed a visible-light-promoted, versatile, tunable, and radical-mediated strategy for the selective C(sp²)–H chlorination and bromination of quinones, (hetero)arenes, natural products and drugs. This practical method is enabled by the in situ generation of the reactive chlorinating reagent, nitrosyl halide, from NaNO$_2$ and aqueous HX. Significantly, the use of commercially available and inexpensive reagents, combined with the mild reaction conditions, broad substrate scope, scalability, and excellent functional group tolerance, renders this protocol particularly useful for the rapid and direct construction of halogenated new drug candidates and highly competitive for the late-stage halogenation of complex bioactive molecules.

## Methods

Selective C(sp²)–H chlorination: Substrate (0.2 mmol, 1.0 equiv), NaNO$_2$ (0.2 mmol, 1.0 equiv), and 37% aqueous HCl (100 μL, 6.0 equiv) were dispersed in DCM (2.0 mL) in a 4 mL glass vial. The vial was sealed with a polytetrafluoroethylene (PTEF) cap, and the reaction mixture was vigorously stirred at room temperature (24 °C) for 10 h under irradiation with 24 W blue LEDs (460–470 nm) positioned 5 cm from the vial. The reaction mixture was then extracted with DCM (3 × 2 mL). The combined organic layers were dried over anhydrous Na$_2$SO$_4$ and filtered, and the filtrate was concentrated. The desired product was purified by chromatography on silica gel.

Selective C(sp²)–H bromination: Substrate (0.2 mmol, 1.0 equiv), NaNO$_2$ (0.2 mmol, 1.0 equiv), and 40% aqueous HBr (100 μL, 3.5 equiv) were dispersed in DCM (2.0 mL) in a 4 mL glass vial. The vial was sealed

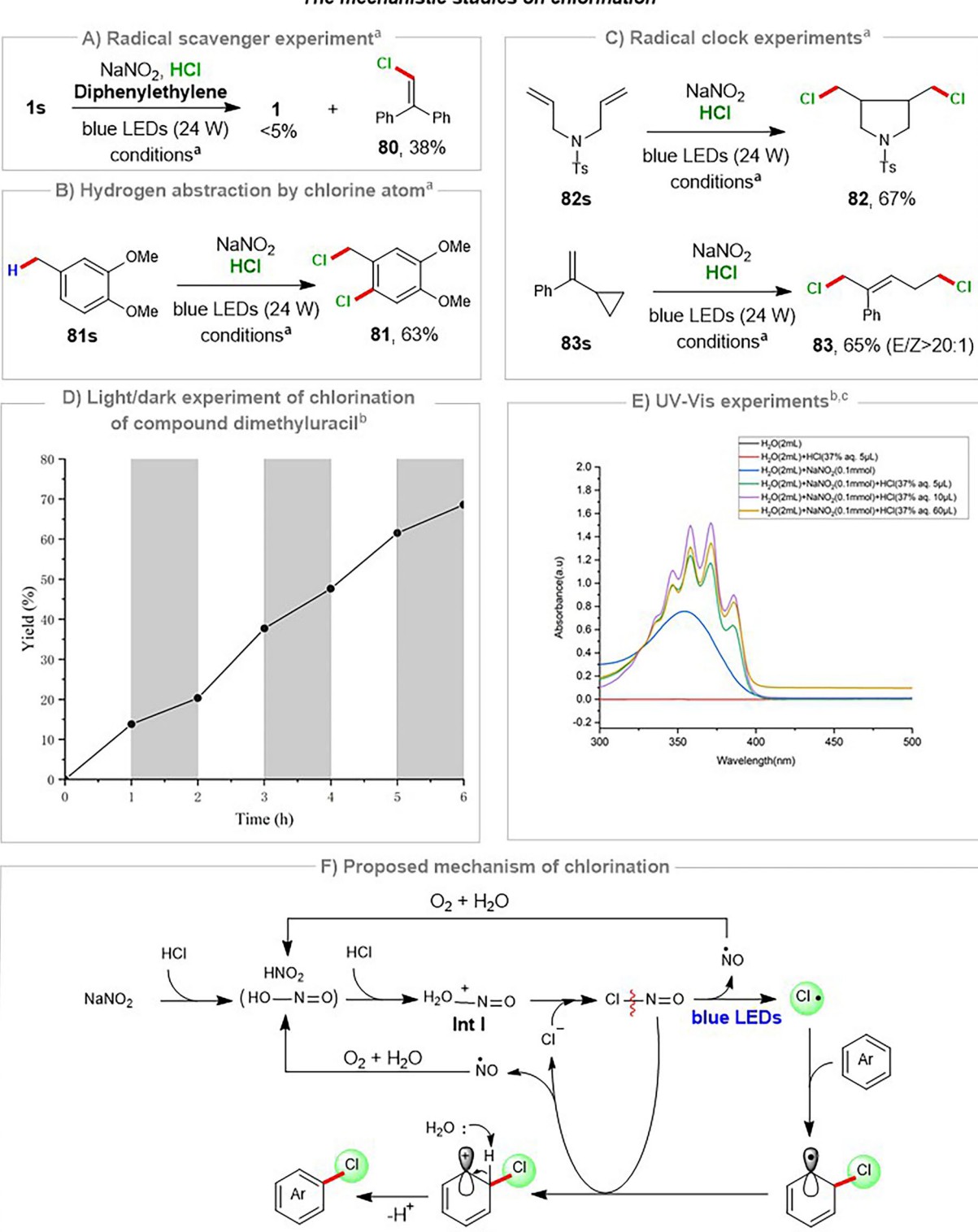

**Fig. 3 | Mechanistic studies. A** Radical scavenger experiment of the chlorination. **B** Hydrogen abstraction of by chlorine atom. **C** Radical clock experiments of the chlorination. **D** Light/dark experiment of chlorination of dimethyluracil. **E** UV–vis experiments. **F** Proposed mechanism of chlorination. ([a]Reaction conditions: substrate (0.2 mmol, 1.0 equiv), NaNO$_2$ (0.2 mmol, 1.0 equiv), and 37% aqueous HCl (100 μL, 6.0 equiv) in DCM (2 mL) were allowed to react at room temperature (24 °C) in air under irradiation by 24 W blue LEDs (460–470 nm). [b]See Supporting Information for more details (S39–42). [c]Experimental UV–vis spectra of 37% aqueous HCl, aqueous NaNO$_2$, and mixtures of 37% aqueous HCl and NaNO$_2$).

**Article**

with a PTEF cap, and the reaction mixture was vigorously stirred at room temperature (24 °C) for 10 h under 24 W blue LEDs (460–470 nm) positioned 5 cm from the vial. The reaction mixture was then extracted with DCM (3 × 2 mL). The combined organic layers were dried over anhydrous $Na_2SO_4$ and filtered, and the filtrate was concentrated. The desired product was purified by chromatography on silica gel.

## Data availability

All data are available within the paper and the Supplementary Information. All methods, figures, tables, schemes, and references in the paper and Supplementary Information are included. NMR spectra are provided as Supplementary Data 1, source data of Table 1 are provided as Supplementary Data 2, and source data of Light/dark experiments are provided as Supplementary Data 3. All data are also available upon request from the corresponding authors.

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

## Acknowledgements

The authors are grateful to the National Science Foundation of China (no. 22371128) and the Qinglan project of Jiangsu province for financial support of this research.

## Author contributions

Y.Y. carried out most of the reactions, evaluated the substrate scope, determined the product structures, and prepared the supporting information. Y.Y. and J.N. carried out mechanism studies. Y.W. formulated the initial concept for this work, supervised the project and prepared the manuscript.

## Competing interests

This work was funded by the National Natural Science Foundation of China [Grant Number 22371128] and the Qinglan project of Jiangsu province. Beyond these research grants, the corresponding author and all other authors declare no other financial or non-financial competing interests.
