## [Transparent Peer Review file · Communications Chemistry]

Visible-light promoted late-stage chlorination and bromination of quinones and (hetero)arenes utilizing aqueous HCl or HBr as halogen donor

Corresponding Author: Professor Yaxin Wang

Version 0:

Reviewer comments:

Reviewer #1

(Remarks to the Author)

The authors have described a method for halogenation of arenes and heteroarenes using HX as halogen source, visible-light activation, and stoichiometric NaNO₂. The system was optimized and demonstrated on a range of quinones and various aromatic and heteroaromatic substrates. It is notable that several examples of late-stage halogenation were included, indicating that the method is quite tolerant of functionality and relatively specific to certain molecular locations. The method is presented as a complementary alternative to other halogenation methods and a majority of the halogenated products have been reported previously. The experiments were conducted with thoughtful detail and scientific rigor, and the characterization of products and description of experimental procedures is very good and detailed enough to be reproduced. There is some concern that the method is not that different from Fukuzumi or König's work using HX and blue LEDs, but this work is photocatalyst-free and has been utilized on relatively complex substrates. Therefore, I believe that publication of the manuscript in Communications Chemistry is warranted in large part based upon the strength of the mechanistic investigation and the inclusion of pharmaceuticals as substrates. Prior to publication, I believe that the following minor revisions need to be addressed or included before finalizing the manuscript:

1) The use of "LED" is inconsistent in some figures and the narrative. In Table 1, Scheme 1, and Scheme 2 it is shown as "Led", and it should be "LED". In addition, there are typos in Figure 3 ("Bule leds" should be "Blue LEDs" in boxes A, B, and C; "Compound" is misspelled in box D). The Table on page 7 of the supporting information (SI) and a few various schemes in the SI also needs to be consistent with regard to "LED".

2) The background seems incomplete with regard to the methods for halogenation using visible-light promotion. The authors have chosen to narrow the photocatalytic halogenation of arenes to systems that employ HX. However, a quick scan of recent review articles that describe photocatalytic halogenation of arenes (Parisotto, S.; Azzi, E.; Lanfranco, A.; Renzi, P.; Deagostino, A., Recent Progresses in the Preparation of Chlorinated Molecules: Electrocatalysis and Photoredox Catalysis in the Spotlight. *Reactions* 2022, 3 (2), 233-253.; and Bellotti, P.; Huang, H. M.; Faber, T.; Glorius, F., Photocatalytic Late-Stage C-H Functionalization. *Chemical reviews* 2023, 123 (8), 4237-4352. - beginning on page 4280) make note of photocatalytic systems that use electrophilic halogenating agents such as succinimides (NXS) under photocatalytic conditions. Since these reactions that employ NXS include some examples of photocatalyzed late-stage halogenation and are closely related (and less atom-economical than HX), I think that they should be included in the background as comparable examples to give a clearer picture of the significance of using HX as halogen source. The articles that seem to be commonly cited are A) Hering, T.; König, B. *Tetrahedron* 2016, 72, 7821; B) Rogers, D. A.; Brown, R. G.; Brandeburg, Z. C.; Ko, E. Y.; Hopkins, M. D.; LeBlanc, G.; Lamar, A. A. *ACS Omega* 2018, 3, 12868; C) Rogers, D. A.; Gallegos, J. M.; Hopkins, M. D.; Lignieres, A. A.; Pitzel, A. K.; Lamar, A. A. *Tetrahedron* 2019, 75, 130498; and D) Rogers, D. A.; Bensalah, A. T.; Espinosa, A. T.; Hoerr, J. L.; Refai, F. H.; Pitzel, A. K.; Alvarado, J. J.; Lamar, A. A. *Organic letters* 2019, 21, 4229. There are some examples of photocatalyzed fluorination as well, but those might be outside of the scope of comparable methods to this article.

3) The mechanistic investigation is a strength of the article, and the results convincingly lead to the implication of a radical species responsible for halogenation. The only experiment that I feel warrants further explanation is the "Light/dark experiment" shown in Figure 3, box D. The observation that the reaction proceeds in the dark indicates that it is more

complicated than a straightforward light-activated reaction and likely involves some form of light-initiated chain propagation in the reaction mechanism (see discussion beginning on page 27242 of Alabugin, I. V.; Eckhardt, P.; Christopher, K. M.; Opatz, T., The Photoredox Paradox: Electron and Hole Upconversion as the Hidden Secrets of Photoredox Catalysis. *Journal of the American Chemical Society* 2024, 146 (40), 27233-27254). An explanation of this result or comment on this possibility would be useful to include.

Upon incorporation of these minor revisions, I believe that this work will provide a valuable addition to the methods available for halogenation of arenes under visible-light-promoted conditions using readily available halogen sources and support its acceptance to *Communications Chemistry*.

Reviewer #2

(Remarks to the Author)

In the presented manuscript, Wang and co-workers demonstrate a process for the chlorination and bromination of arenes via the in-situ generation and cleavage of NOCl and NOBr in the presence of light. While this serves as an alternate method of halide insertion, the substrate scope is limited to electron-rich systems, similar to the work demonstrated by König and co-workers (as cited in the text) as well as others (Ref. 13). While the method developed is certainly interesting, the existence of similar, and even better processes, slightly lowers the impact of this work when seen in terms of publication in *Communications Chemistry*. As a result, this reviewer is of the opinion that the work may be published elsewhere. Before submission in other journals, the authors should try pondering over the following concerns:

1. The authors have explored only electron rich arenes and hetero-arenes throughout the substrate scope. What is the fate of the reaction when substrates such as benzene, or electron-poor arenes are used?
2. The authors should explore the outcome of the reaction when phenols are used. The results should be clearly mentioned in the manuscript.
3. For substrate 16, is the p-chlorination exclusive? Rationally, o-chlorination may also be possible. Authors should clarify this result.
4. For several substrates, di-chloro- and di-bromination is observed. The authors should clarify if this di-halogenation is uncontrolled. If there is any possibility of control, the authors should pursue a selective mono-halogenation.
5. The authors should mention the fate of the reaction when oxygen purging is done.

Reviewer #3

(Remarks to the Author)

The work makes a good impression. The method proposed by the authors is a soft alternative to the use of elemental chlorine and bromine for halogenation of aromatic compounds (in particular, see products 82, 83). But unlike them, in situ generated nitrosyl chloride is easier to handle and demonstrates higher selectivity. In terms of procedure simplicity, the proposed conditions are close to ideal. Not very high excesses of available reagents and testing of flow setups also suggest high potential for the pharmaceutical industry. Overall, this work is suitable for publication in *Commun Chem* after minor revision.

Nevertheless, I have a couple of comments.

1. A comparison of selectivity of halogenation with that of several known methods (in particular, halogenation with elemental halogens), at least using 1-2 substrates as an example, should be provided. It is also worth citing some related works: *J. Am. Chem. Soc.* 2022, 144, 2399–2414; *Nat. Chem.* 2024, 16, 1539–1545.
2. It is worth changing the format of Table 1: instead of the volume of the HCl solution, it is better to indicate equivalents - this will help the reader. For example, instead of "NaNO₂ (1.0), HCl (50)" it is worth writing "NaNO₂ (1.0), HCl (6.0)".

Version 1:

Reviewer comments:

Reviewer #1

(Remarks to the Author)

After reviewing the revisions submitted by the authors, I believe that the work is ready for publication.

Reviewer #2

(Remarks to the Author)

In the revised manuscript, the authors have effectively addressed most concerns regarding the novelty of their research compared to earlier studies. They have also clearly outlined the limitations related to substrate scope and addressed the issue of controlling selective mono- and di-halogenation products. After the revision, the novelty of the work has been

significantly enhanced, and the manuscript now meets the standards of this journal. As a result, the current manuscript appears suitable for publication in Communications Chemistry.

Reviewer #3

(Remarks to the Author)

All my comments have been addressed. The manuscript can be accepted in the current state.

Response to the reviewers

Reviewers' comments:

Reviewer #1 (Remarks to the Author):

The authors have described a method for halogenation of arenes and heteroarenes using HX as halogen source, visible-light activation, and stoichiometric NaNO₂. The system was optimized and demonstrated on a range of quinones and various aromatic and heteroaromatic substrates. It is notable that several examples of late-stage halogenation were included, indicating that the method is quite tolerant of functionality and relatively specific to certain molecular locations. The method is presented as a complementary alternative to other halogenation methods and a majority of the halogenated products have been reported previously. The experiments were conducted with thoughtful detail and scientific rigor, and the characterization of products and description of experimental procedures is very good and detailed enough to be reproduced. There is some concern that the method is not that different from Fukuzumi or Konig's work using HX and blue LEDs, but this work is photocatalyst-free and has been utilized on relatively complex substrates. Therefore, I believe that publication of the manuscript in Communications Chemistry is warranted in large part based upon the strength of the mechanistic investigation and the inclusion of pharmaceuticals as substrates. Prior to publication, I believe that the following minor revisions need to be addressed or included before finalizing the manuscript:

Response:

We are grateful for referee's positive comments and the suggestion to publish our work in Communications Chemistry. We have carefully considered your valuable comments and have made the necessary revisions to our manuscript and supporting information accordingly to enhance the quality and clarity of our work. We believe that these revisions have strengthened our manuscript and brought it in line with the standards of Communications Chemistry.

1) The use of "LED" is inconsistent in some figures and the narrative. In Table 1, Scheme 1, and Scheme 2 it is shown as "Led", and it should be "LED". In addition, there are typos in Figure 3 ("Bule leds" should be "Blue LEDs" in boxes A, B, and C; "Compound" is misspelled in box D). The Table on page 7 of the supporting information (SI) and a few various schemes in the SI also needs to be consistent with regard to "LED".

Response:

Thank you very much for carefully examining our manuscript and for pointing out the

inconsistent use of “LED”. We have revised every sentence, figure, and scheme involving “LED” to ensure consistent usage, and the changes are highlighted in yellow in the manuscript.

2) The background seems incomplete with regard to the methods for halogenation using visible-light promotion. The authors have chosen to narrow the photocatalytic halogenation of arenes to systems that employ HX. However, a quick scan of recent review articles that describe photocatalytic halogenation of arenes (Parisotto, S.; Azzi, E.; Lanfranco, A.; Renzi, P.; Deagostino, A., Recent Progresses in the Preparation of Chlorinated Molecules: Electrocatalysis and Photoredox Catalysis in the Spotlight. *Reactions* 2022, 3 (2), 233-253.; and Bellotti, P.; Huang, H. M.; Faber, T.; Glorius, F., Photocatalytic Late-Stage C-H Functionalization. *Chemical reviews* 2023, 123 (8), 4237-4352. - beginning on page 4280) make note of photocatalytic systems that use electrophilic halogenating agents such as succinimides (NXS) under photocatalytic conditions. Since these reactions that employ NXS include some examples of photocatalyzed late-stage halogenation and are closely related (and less atom-economical than HX), I think that they should be included in the background as comparable examples to give a clearer picture of the significance of using HX as halogen source. The articles that seem to be commonly cited are A) Hering, T.; König, B. *Tetrahedron* 2016, 72, 7821; B) Rogers, D. A.; Brown, R. G.; Brandeburg, Z. C.; Ko, E. Y.; Hopkins, M. D.; LeBlanc, G.; Lamar, A. A. *ACS Omega* 2018, 3, 12868; C) Rogers, D. A.; Gallegos, J. M.; Hopkins, M. D.; Lignieres, A. A.; Pitzel, A. K.; Lamar, A. A. *Tetrahedron* 2019, 75, 130498; and D) Rogers, D. A.; Bensalah, A. T.; Espinosa, A. T.; Hoerr, J. L.; Refai, F. H.; Pitzel, A. K.; Alvarado, J. J.; Lamar, A. A. *Organic letters* 2019, 21, 4229. There are some examples of photocatalyzed fluorination as well, but those might be outside of the scope of comparable methods to this article.

Response:

We thank the reviewer for this constructive suggestion and for pointing out the two authoritative reviews (*Reactions* 2022, 3, 233; *Chem. Rev.* 2023, 123, 4280). Upon re-examining the literature, we fully agree with the reviewer’s suggestion. In the revised manuscript we have systematically categorized the visible-light-promoted aromatic C(sp²)-H halogenation methods and incorporated them into the Introduction (see the third paragraph of the revised manuscript). A concise comparative figure has also been added to highlight the advantages of HX-based protocols over the NXS-mediated approaches (see Figure 1C, in the revised manuscript). Furthermore, we have added these important references on the aromatic C(sp²)-H halogenation reactions to our manuscript.

The changes have been highlighted in yellow in the revised manuscript. We believe that these revisions and references provide valuable context and support for our research, and we appreciate your suggestion to include them.

3) The mechanistic investigation is a strength of the article, and the results convincingly lead to the implication of a radical species responsible for halogenation. The only experiment that I feel warrants further explanation is the "Light/dark experiment" shown in Figure 3, box D. The observation that the reaction proceeds in the dark indicates that it is more complicated than a straightforward light-activated reaction and likely involves some form of light-initiated chain propagation in the reaction mechanism (see discussion beginning on page 27242 of Alabugin, I. V.; Eckhardt, P.; Christopher, K. M.; Opatz, T., *The Photoredox Paradox: Electron and Hole Upconversion as the Hidden Secrets of Photoredox Catalysis*. *Journal of the American Chemical Society* 2024, 146 (40), 27233-27254). An explanation of this result or comment on this possibility would be useful to include.

Response:

We sincerely thank the reviewer for providing the important reference (*J. Am. Chem. Soc.* 2024, 146, 27233) to further explain the observation that the reaction proceeds in the dark.

Guided by the reviewer's suggestion and the literature mentioned by the reviewer, we now propose that a radical-chain pathway is involved after illumination ceases, and the corresponding mechanistic discussion, proposed mechanism and reference mentioned by the reviewer has been incorporated into both the revised manuscript and the supporting Information.

The clear self-sustained component points toward a radical chain process being involved (Figure R1). Firstly, the chlorine radical generated through the photolysis of nitrosyl chloride react with the π -system of the arene to afford the corresponding carbon-centered radical intermediate (**Int II**). **Int II** undergoes a single-electron transfer (SET) with nitrosyl chloride to generate carbocation intermediate (**Int III**). A trace amount of H₂O present in aqueous HCl then acts as a base to facilitate proton elimination of **Int III**, furnishing the aromatic C(sp²)-H chlorinated product. Meanwhile, the one-electron-reduced nitrosyl chloride fragments to afford a nitric oxide anion (NO⁻) and a chlorine radical (Cl•); the regenerated Cl• then propagates the radical chain. Thermodynamically, the cleavage of the one-electron-reduced nitrosyl chloride to generate nitric oxide anion (NO⁻) and a chlorine radical (Cl•) is highly disfavoured; nevertheless, rapid downstream consumption of the nascent Cl• would shift the equilibrium toward chloride formation, rendering the fragmentation kinetically feasible.

The corresponding content have been added in the revised supporting Information (page 41-43).

Figure R1. Proposed radical-chain mechanism of C(sp²)-H chlorination

Upon incorporation of these minor revisions, I believe that this work will provide a valuable addition to the methods available for halogenation of arenes under visible-light-promoted conditions using readily available halogen sources and support its acceptance to Communications Chemistry.

Response:

We thank the reviewer for this supportive summary. All revisions suggested by the reviewer have been carefully incorporated into the revised manuscript and the supporting Information (the changes are highlighted in yellow in both the manuscript and the supporting Information). We trust that these changes satisfactorily address the reviewer's concerns. Thank you again for helping us improve this work.

Reviewer #2 (Remarks to the Author):

In the presented manuscript, Wang and co-workers demonstrate a process for the chlorination and bromination of arenes via the in-situ generation and cleavage of NOCl and NOBr in the presence of light. While this serves as an alternate method of halide insertion, the substrate scope is limited to electron-rich systems, similar to the work demonstrated by König and co-workers (as cited in the text) as well as others (Ref. 13). While the method developed is certainly interesting, the existence of similar, and even better processes, slightly lowers the impact of this work when seen in terms of publication in Communications Chemistry. As a result, this reviewer is of the opinion that the work may be published elsewhere. Before submission in other journals, the authors should try pondering over the following concerns:

Response:

We thank the reviewer for raising this point. We have carefully compared our work with the halogenation protocols developed by the König group and others, and a side-by-side analysis reveals fundamental differences in reaction manifold, scope, and outcome.

Angew. Chem. Int. Ed. 2016, 55, 5342–5345 –reports the photo-promoted aromatic C(sp²)-H chlorination of electron-rich arenes using flavin photocatalysis. Firstly, compared with the reported enzymatic system of König group, our synthetic methodology operates under entirely different principles and conditions. Moreover, the substrate scope described in the reference is confined to highly activated, simple, low-molecular-weight electron-rich arenes and exhibits narrow functional-group tolerance. The authors also note that, because the enzymes are naturally substrate-specific, the range of accessible products is inherently limited. In contrast, our protocol demonstrates exceptional compatibility with a broad array of electron-rich arenes, including those bearing sensitive functional groups. Even electron-deficient arenes or aza-arenes carrying electron-donating groups (such as 2,6-dimethoxybenzoic acid and 2,6-dimethoxypyridine) also undergo selective C–H chlorination under our conditions. Most importantly, our system enables late-stage halogenation of complex drug molecules and natural products, applications that are completely absent from the enzymatic dataset. Consequently, the two studies address non-overlapping chemical spaces.

Eur. J. Org. Chem. 2020, 1491–1495 –describes oxidative photochlorination of electron-rich arenes via in situ bromination. In this system, electron-rich arenes are oxidatively photochlorinated in the presence of catalytic amounts of bromide ions, visible light, and organic photoredox catalyst 4CzIPN. The substrates are brominated *in situ* in a first photoredox-catalyzed oxidation step, followed by a photocatalyzed ipso-chlorination, yielding selective C(sp²)-H chlorinated product. Our approach

utilizes sodium nitrite and aqueous hydrochloric acid under visible-light irradiation to accomplish without any photocatalyst. Moreover, our system is highly tunable: simple replacement of aqueous HCl with aqueous HBr delivers selective aromatic C(sp²)-H bromination under otherwise identical conditions. Furthermore, the substrate scope reported by König group is limited to simple and low-molecular-weight electron-rich arenes, whereas our system demonstrates outstanding compatibility with a diverse array of arenes and heteroarenes. Therefore, the two systems are non-overlapping in both mechanism and scope.

Tetrahedron 2016, 72, 7821-7825 –reports visible-light photoredox activation of N-chloramines NCS for electrophilic aromatic chlorination. This protocol relies on the expensive photocatalyst Ru(bpy)₃Cl₂ and stoichiometric oxidant (NH₄)₂S₂O₈. In contrast, our methodology operates under entirely different mechanistic principles and avoids costly photocatalyst and external oxidants. Moreover, the cited substrate scope is restricted to simple, low-molecular-weight electron-rich arenes, whereas our system exhibits exceptional compatibility with diverse (hetero)arenes bearing various functional groups. Importantly, it enables late-stage halogenation of complex drugs and natural products—applications absent from the previous dataset. Consequently, the two studies address non-overlapping chemical spaces.

To date, visible-light-promoted C(sp²)-H halogenation of (hetero)arenes has been dominated by two paradigms. The first relies on N-Cl or N-Br reagents in combination with a photocatalyst to deliver selective aromatic C(sp²)-H chlorinated or brominated products. This manifold requires a photocatalyst and inevitably generates stoichiometric by-products, thereby compromising the atom economy. The second manifold employs aqueous HCl or HBr, or inorganic halide salts, as the halogen source; a strong chemical oxidant or strongly oxidizing photocatalyst is then required to generate the corresponding halogen radical that adds to the (hetero)arene. These strongly oxidative conditions erode functional-group tolerance and restrict substrate scope, rendering late-stage C(sp²)-H halogenation of complex natural products or drug scaffolds difficult.

Our protocol distinguishes itself in key aspects:

(i) Reaction system and conditions:

Our method achieves selective aromatic C(sp²)-H chlorination of (hetero)arenes using inexpensive, low-molecular-weight aqueous HCl and commercially available sodium nitrite under blue-LED irradiation, without any photocatalyst, metal catalyst, or stoichiometric oxidant. Moreover, the protocol can be readily extended to C(sp²)-H bromination by simply replacing 37 % aqueous HCl with 40 % aqueous HBr. Overall, the system is practical, low-cost, highly tunable, and selective.

(ii) Substrate scope and Applications:

The substrate scope of chlorination works mentioned by reviewer is limited to simple and low-molecular-weight electron-rich (hetero)arenes. Compared with these

aromatic C(sp²)-H chlorination reactions, our system exhibits exceptional compatibility with diverse quinones, arenes, and heteroarenes bearing a broad range of functional groups. Particularly, our system is useful for the late-stage C(sp²)-H halogenation of complex drug molecules and natural products, which will aid in the construction of new drug candidates.

(iii) Practicality:

Compared with previously visible-light-promoted aromatic C(sp²)-H halogenation reactions, our system shows versatility, practicality, operational simplicity, high selectivity and efficiency, employs low-cost and commercially available reagents under mild conditions, and allows easy scale-up via continuous-flow reaction technology.

In the revised manuscript we have systematically categorized the visible-light-promoted aromatic C(sp²)-H halogenation methods and incorporated them into the Introduction (see the third paragraph of the revised manuscript and Figure 1). A concise comparative figure has also been added to highlight the advantages of our reaction system over previous protocols. We hope these additions adequately address the reviewer's novelty concern.

1. The authors have explored only electron rich arenes and hetero-arenes throughout the substrate scope. What is the fate of the reaction when substrates such as benzene, or electron-poor arenes are used?

Response:

We thank the referee for the insightful suggestion. To address your concern we have now carried out additional experiments with benzene and a set of representative electron-deficient arenes under our standard conditions. The new results are summarized in Scheme R1 and pages 32–34 of the revised supporting information.

Key observations:

- 1) Electron-poor arenes and hetero-arenes such as benzoic acid, nitrobenzene, pyridine, quinolone, and isoquinoline were completely unreactive under our standard chlorination conditions (Scheme R1, A).
- 2) When the electron-deficient heterocycle carries electron-donating groups, e.g., 2,6-dimethoxypyridine, the chlorination can proceed to give the corresponding selective chlorinated product in 34% yield (Scheme R1, A).
- 3) Benzene itself gave 46% mono-chlorination product under our standard chlorinated conditions (Scheme R1, A).
- 4) Electron-poor arenes and hetero-arenes such as benzoic acid, nitrobenzene, pyridine, quinolone, and isoquinoline were completely unreactive under our standard bromination conditions (Scheme R1, B).
- 5) When the electron-deficient heteroarenes or arenes carry electron-donating groups, e.g., 2,6-dimethoxypyridine, 2,6-dimethoxybenzoic acid or

3,4-dimethoxybenzaldehyde, the bromination can proceed to give the corresponding selective brominated products in moderate to good yields (Scheme R1, B).

- 6) Benzene itself gave 60% mono-bromination product under our standard brominated conditions (Scheme R1, B).

The corresponding content have been added in the revised supporting Information (page 33-34). We hope that the additional experiments and discussion satisfactorily clarify the substrate-scope limitation. We are fully aware that electron-poor (hetero)arenes are challenging substrates under the current protocol. Ongoing work in our laboratory is focused on extending the present chlorination/bromination to electron-deficient arenes and heteroarenes; these results will be reported in due course.

A) Visible-light-promoted C(sp²)-H chlorination of electron-poor arenes and benzene

B) Visible-light-promoted C(sp²)-H bromination of electron-poor arenes and benzene

Scheme R1. Visible-light-promoted C(sp²)-H chlorination and bromination of electron-poor arenes and benzene

2. The authors should explore the outcome of the reaction when phenols are used. The results should be clearly mentioned in the manuscript.

Response:

We thank the reviewer for this helpful suggestion.

Following your suggestion, we examined the reaction with phenol as the substrate.

Treating phenol under the standard visible-light-promoted, NaNO₂/HCl-mediated conditions gave a complex mixture; no chlorophenol was detected. ¹H NMR and GC-MS of the crude reaction showed that phenol was largely oxidized to *p*-benzoquinone or *o*-benzoquinone. Because the chlorine radical is a stronger oxidant, phenols are preferentially oxidized to quinone derivatives.

Treating phenol under the standard visible-light promoted NaNO₂/HBr mediated C–H bromination conditions gave 4-bromophenol in 78 % isolated yield (no *ortho*-isomer observed by ¹H-NMR and GC-MS).

These results have been added to the revised manuscript (page 10, Scheme 2, compound **56'**) and the experimental details are provided in the supporting information (Page 25-26).

3. For substrate 16, is the *p*-chlorination exclusive? Rationally, *o*-chlorination may also be possible. Authors should clarify this result.

Response:

We thank the reviewer for raising this point. For substrate 16, the *para*-chlorinated product is indeed formed exclusively (no *ortho*-isomer detected by ¹H-NMR and GC-MS of the crude reaction mixture). Only the *para*-chlorinated product was observed, presumably because the *ortho*-positions are sterically hindered. We also note that previous C(sp²)-H chlorinations of alkoxybenzenes under electrophilic conditions have uniformly delivered only the *para*-isomer (see, for example, refs: Angew. Chem. Int. Ed. 2023, 62, e20231259 (compound 7a) and Nature Catalysis, 2020, 3, 107-115. (compound Cl-44)), in full agreement with our result.

4. For several substrates, di-chloro- and di-bromination is observed. The authors should clarify if this di-halogenation is uncontrolled. If there is any possibility of control, the authors should pursue a selective mono-halogenation.

Response:

We thank the reviewer for pointing out the important issue of di-halogenation. Below we clarify (i) whether the di-halogenation is “uncontrolled” and (ii) what we have done to achieve selective mono-halogenation.

In the chlorination of 1,4-benzoquinone, our standard chlorination conditions yielded 56% of the mono-chlorinated and 32% of the di-chlorinated products (**3** and **3'**). However, by precisely reducing the aqueous HCl loading to 3.0 equivalents, the mono-chlorinated benzoquinone (**3**) was obtained as the sole isolable product in 87% isolated yield. Using a similar strategy, we attempted to obtain the mono-chlorinated product in the chlorination of anthracene-1,4-dione; however, only a trace amount of the mono-chlorinated anthracene-1,4-dione was formed (~3%), and the majority of the starting material was recovered (the recovery of starting material anthracene-1,4-dione is 90%).

In the bromination of 1,4-dimethoxybenzene and 1,3-dimethoxybenzene, our standard conditions afforded the dibrominated products **51** and **52** in excellent yields. However, by precisely reducing the aqueous HBr loading to 1.7 equivalents, both substrates provided the monobrominated product in 90% yield. Likewise, by reducing the aqueous HBr we sought to obtain the mono-brominated 2,6-dimethoxybenzoic acid and 2,6-dimethoxypyridine; regrettably, only a trace amount of the mono-brominated products were formed, and the majority of the starting material was recovered (the recovery of starting materials are 90%).

The relevant data have been summarized in Scheme 1 and the corresponding discussion can be found on page 7-8 in the revised manuscript (The changes have been highlighted in yellow in the revised manuscript). These experimental details are provided in the supporting information (Page 9 and 24).

We believe that these revisions provide valuable context and support for our research, and we appreciate your suggestion to include them.

5. The authors should mention the fate of the reaction when oxygen purging is done.

Response:

We sincerely thank the reviewer for this insightful comment. To evaluate the potential role of atmospheric oxygen, we performed the selective aromatic C(sp²)-H chlorination of naproxen (**1s**) under rigorously controlled atmospheres (air, O₂, and N₂). All reactions were run in triplicate, and conversions and yields were determined by ¹H NMR analysis of the crude reaction mixture or by column chromatography (Table R1).

Entry	Atmosphere	Yield 1 (%)	Yield 1' (%)	Recovery of starting material 1s
1	air	85(75) ^b	<10	0
2	O₂	67(60) ^b	18(10) ^b	0
3	N₂	17(8) ^b	<10	70

^aStandard reaction conditions: naproxen (0.1 mmol, 1.0 equiv), NaNO₂ (0.1 mmol, 1.0 equiv), and 37% aqueous HCl (50 uL, 6.0 equiv) in DCM (1 mL) were allowed to react at room temperature (24 °C) in air under irradiation with 24 W blue LEDs (460-470 nm). The yields are ¹H-NMR yields. ^bThe yields are isolated yields.

Table R1. C(sp²)-H chlorination of naproxen under different atmosphere

The reaction run under ambient air gave the best yield of the C(sp²)-H chlorination product **1** (85 %). Under an O₂-rich atmosphere, the chlorinated product **1** was obtained in 67 % yield, together with the nitration product **1'** in 18 % yield. In contrast, under a N₂ atmosphere only 17 % of **1** was formed and 70 % of the starting material remained unreacted. These results demonstrate that ambient air exerts a beneficial effect on the reaction, whereas an excess of O₂ rapidly oxidizes NO to NO₂, diverting the pathway toward nitration and increasing the yield of the nitro product **1'**. Conversely, under an inert N₂ atmosphere the transformation is markedly retarded, underscoring the pivotal role of molecular oxygen in regenerating the chlorinating reagent ClNO and sustaining the radical process (see mechanism in the revised manuscript, Figure 3).

The relevant data have been summarized in Table 1 (entries 4, 10 and 11) and the corresponding discussion can be found on page 5, lines 24–35. The changes have been highlighted in yellow in the revised manuscript. We believe that these revisions provide valuable context and support for our research, and we appreciate your suggestion to include them.

Reviewer #3 (Remarks to the Author):

The work makes a good impression. The method proposed by the authors is a soft alternative to the use of elemental chlorine and bromine for halogenation of aromatic compounds (in particular, see products 82, 83). But unlike them, in situ generated nitrosyl chloride is easier to handle and demonstrates higher selectivity. In terms of procedure simplicity, the proposed conditions are close to ideal. Not very high excesses of available reagents and testing of flow setups also suggest high potential for the pharmaceutical industry. Overall, this work is suitable for publication in *Commun Chem* after minor revision.

Response:

We are grateful for referee's positive comments and the suggestion to publish our work in *Communications Chemistry*. We have carefully considered your valuable comments and have made the necessary revisions to our manuscript and supporting information accordingly to enhance the quality and clarity of our work. We believe that these revisions have strengthened our manuscript and brought it in line with the high standards of *Communications Chemistry*.

Nevertheless, I have a couple of comments.

1. A comparison of selectivity of halogenation with that of several known methods (in particular, halogenation with elemental halogens), at least using 1-2 substrates as an example, should be provided. It is also worth citing some related works: *J. Am. Chem. Soc.* 2022, 144, 2399–2414; *Nat. Chem.* 2024, 16, 1539–1545.

Response:

We thank the reviewer for this helpful suggestion. Following the recommendation, we used the C(sp²)-H chlorination of naproxen as a test case and compared our NaNO₂/HCl-mediated visible-light protocol with visible-light-promoted chlorination using NCS, DCDMH, PhICl₂ or Palau'chlor (Table 1, entries 12–15). These reagents afforded the target product in 10 %, 60 %, 68 % and 56 % yield, respectively, whereas our NaNO₂/HCl system delivered 85 % yield under the same irradiation conditions. The data are summarized in Table 1 and discussed on page 5, lines 35–38 of the revised manuscript. Moreover, we have added the important references suggested by the reviewer to the manuscript.

The changes have been highlighted in yellow in the revised manuscript. We believe that these revisions and references provide valuable context and support for our research, and we appreciate your suggestion to include them.

2. It is worth changing the format of Table 1: instead of the volume of the HCl solution, it is better to indicate equivalents - this will help the reader. For example, instead of "NaNO₂ (1.0), HCl (50)" it is worth writing "NaNO₂ (1.0), HCl (6.0)".

Response:

Thank you for this helpful suggestion. We agree that reporting equivalents makes the procedure clearer and more reproducible. In the revised manuscript and the supporting Information we have accordingly added the equivalents of HCl. All other stoichiometry-dependent footnotes and the Experimental section have been updated.

Response to the reviewers

Reviewers' comments:

Reviewer #1 (Remarks to the Author):

After reviewing the revisions submitted by the authors, I believe that the work is ready for publication.

Response: We are grateful for referee's positive comments and the suggestion to publish our work in Communications Chemistry.

Reviewer #2 (Remarks to the Author):

In the revised manuscript, the authors have effectively addressed most concerns regarding the novelty of their research compared to earlier studies. They have also clearly outlined the limitations related to substrate scope and addressed the issue of controlling selective mono- and di-halogenation products. After the revision, the novelty of the work has been significantly enhanced, and the manuscript now meets the standards of this journal. As a result, the current manuscript appears suitable for publication in Communications Chemistry.

Response: We are grateful for referee's positive comments and the suggestion to publish our work in Communications Chemistry.

Reviewer #3 (Remarks to the Author):

All my comments have been addressed. The manuscript can be accepted in the current state.

Response: We are grateful for referee's positive comments and the suggestion to publish our work in Communications Chemistry.